# The ML.ENERGY Benchmark: Toward Automated Inference Energy Measurement and Optimization

**Jae-Won Chung    Jeff J. Ma    Ruofan Wu    Jiachen Liu**
**Oh Jun Kweon    Yuxuan Xia    Zhiyu Wu    Mosharaf Chowdhury**

University of Michigan
**The ML.ENERGY Initiative**

## Abstract

As the adoption of Generative AI in real-world services grow explosively, *energy* has emerged as a critical bottleneck resource. However, energy remains a metric that is often overlooked, under-explored, or poorly understood in the context of building ML systems. We present the ML.ENERGY Benchmark, a benchmark suite and tool for measuring inference energy consumption under realistic service environments, and the corresponding ML.ENERGY Leaderboard, which have served as a valuable resource for those hoping to understand and optimize the energy consumption of their generative AI services. In this paper, we explain four key design principles for benchmarking ML energy we have acquired over time, and then describe how they are implemented in the ML.ENERGY Benchmark. We then highlight results from the early 2025 iteration of the benchmark, including energy measurements of 40 widely used model architectures across 6 different tasks, case studies of how ML design choices impact energy consumption, and how automated optimization recommendations can lead to significant (sometimes more than 40%) energy savings without changing what is being computed by the model. The ML.ENERGY Benchmark is open-source and can be easily extended to various customized models and application scenarios.

## 1 Introduction

Generative AI models have rapidly transitioned from research prototypes to real-world services such as ChatGPT [56], Character AI [6], Sora [57], and Midjourney [50]. However, exponential growth rarely continues without facing scaling bottlenecks; currently for generative AI, one of the most crucial bottlenecks is the *energy bottleneck* [4, 15–17, 38, 48, 49, 51]. That is, even with fleets of latest GPUs and exploding demand for ML compute, getting access to the energy necessary to power these systems is becoming increasingly costly, slow, and sometimes impossible. This particularly impacts serving real-world services as ML inference reportedly accounts for 80–90% of the total compute demand [12, 32, 58, 60]. Left unaddressed, the energy bottleneck will not only hinder AI research and development progress [31], but also lead to energy being squeezed out of existing electricity grids and impacting availability and price [4].

However, despite its growing importance, energy remains a secondary consideration compared to traditional optimization objectives like time and accuracy. How much energy does a model consume during inference? What is the right way for energy measurement and accounting during execution, let alone optimization? To bridge this gap, we launched the ML.ENERGY Leaderboard,[1] the first inference energy leaderboard for modern generative AI models to the best of our knowledge. We have

---

[1] https://ml.energy/leaderboard

39th Conference on Neural Information Processing Systems (NeurIPS 2025) Track on Datasets and Benchmarks.

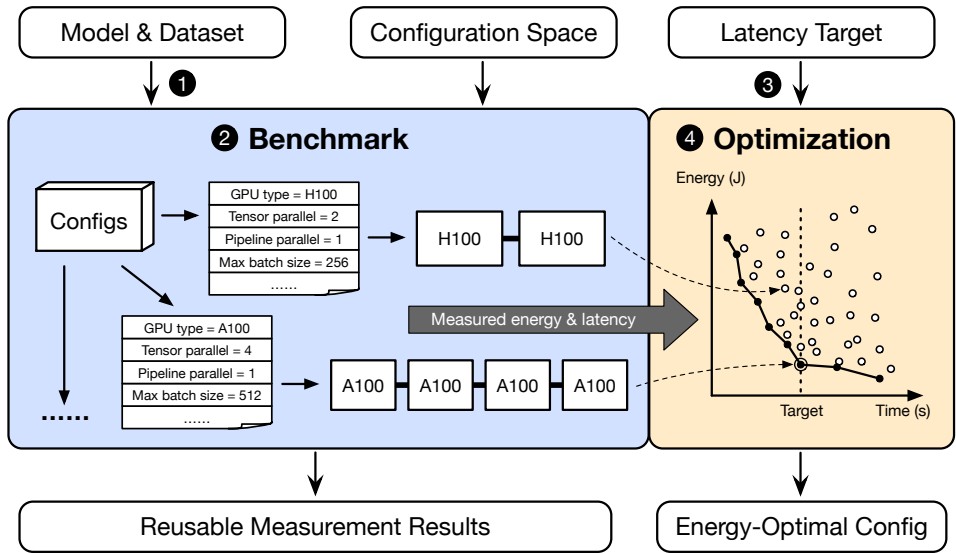

Figure 1: Overview of the benchmarking and optimization flow of the ML.ENERGY Benchmark.

been gradually expanding the Leaderboard in multiple dimensions to now include (1) 40 different generative AI model architectures across a wide range of tasks – including Large Language Model (LLM) chat and coding, Vision–Language Model (VLM) visual chat, and text-to-image, text-to-video, and image-to-video generation using Diffusion models – and (2) more up-to-date hardware and software stacks following rapid advancements in each area.

In this paper, we share the design principles we have established over time (Section 2) and present the ML.ENERGY Benchmark that embodies them (Section 3). It provides two key functionalities:

- **Extensible benchmark**: It provides an easily extensible benchmark suite and a comprehensive set of tools for measuring the inference energy consumption of generative AI models for various tasks under *realistic* deployment environments.
- **Automated optimization**: Based on energy measurement results, it provides automated energy optimization recommendations for generative AI model deployment.

Finally, we highlight notable results from the early 2025 iteration of the ML.ENERGY Leaderboard, shedding light on (1) how energy consumption varies across different generative AI models and tasks, (2) the complex trade-offs that involve energy, time, and model architecture design, and (3) the energy savings opportunity unlocked by automated optimization (Section 4).

This paper describes the state of the ML.ENERGY Benchmark and Leaderboard as of *early 2025*. The latest version of the ML.ENERGY Benchmark is open-source on GitHub,[2] and the ML.ENERGY Leaderboard allows everyone to browse full results from the latest ML.ENERGY Benchmark.

## 2 Design Principles

The design of the ML.ENERGY Benchmark is guided by four core principles. Our overarching goal is to create a benchmark that is representative of real-world generative AI service deployments, and to produce energy measurement results that are accurate, reusable, and ultimately actionable.

### 2.1 Generalizability and Portability

**Goal.** Every computer system is configured with different hardware and software components, and measurements from a particular system will never truly represent those from another system. For instance, systems can be configured with different CPU and DRAM models, and running different

---

[2]https://github.com/ml-energy/benchmark

Linux kernel versions with different daemons running in the background. Further, not all users have physical access to the target system hardware, a common case for cloud-based environments. Still, we wanted (1) the benchmark to run seamlessly on a wide variety of systems, and (2) measurement results to provide generalizable insights and recommendations across a wide range of systems.

**Our approach.**  We focus on software-based GPU energy measurement for the following reasons:

- GPUs are the dominant worker and energy consumer in a system running ML services, accounting for 50–70% of the total provisioned power in the datacenter [52–54, 58].
- Compared to other hardware components, GPU models are more standardized across different systems [13], making measurements useful across systems that use the same GPU.
- GPUs allow accurate software-based energy measurement [1, 2, 11, 81], allowing measurement tools to be portable across systems without requiring physical hardware access or modification.

## 2.2  Representing Real-World Deployments

**Goal.**  Benchmarking results often inform real-world deployment optimizations, are used to plan future power capacity and energy usage, affect the design of new hardware and software systems, and serve as base numbers for long term projections that affect policymaking. Therefore, it is crucial that our measurements represent those from real-world deployments as closely as possible.

**Our approach.**  To obtain realistic measurements, we adhere to the following principles:

- We adopt production-grade software and hardware (e.g., vLLM [39] on NVIDIA H100 GPUs) and run them with generation request workloads that are representative of real-world use cases.
- During our measurement, we directly run or closely mimic the state of a serving system during long term deployment. This allows us to capture the *steady state* energy consumption of the service while using a fixed-size benchmarking dataset.

## 2.3  Energy Measurement at the Right Granularity

**Goal.**  Energy can be measured at different computation granularities. For instance, for LLM text generation, energy can be reported for the end-to-end benchmarking run, for each generated response, or for each token generated. Our goal is to measure and report energy consumption at a granularity that is neither too coarse (as it only provides limited insight into the runtime behavior of the service) nor too fine (as it may miss important higher-level insights relevant to the service).

**Our approach.**  Also aligned with our goal of representing real-world deployments (Section 2.2), our approach is to mainly report energy consumption at the granularity of a single, whole generation response to a request (e.g., entire chat response, image, video). This is because any work less than the full response (e.g., per token) is not considered a complete request, and may ignore model- and task-specific characteristics. For instance, for LLM text generation, different models exhibit different *verbosity* (i.e., given the same prompt, different models respond with varying number of tokens), and different tasks have vastly different output token length distributions (e.g., chat vs. code generation), all of which we want to capture in our measurements.

## 2.4  Actionable Measurement Results

**Goal.**  While energy measurements are useful in themselves, they are even more useful when they lead to actionable insights and recommendations. For instance, how much is the potential energy savings of your model without sacrificing accuracy or latency? If your service intends to guarantee a specific generation latency deadline (e.g., 50 ms), what is the energy-optimal configuration, and how much is the potential energy savings?

**Our approach.**  The ML.ENERGY Benchmark allows users to provide computation latency constraints specific to their application scenario (e.g., LLM average Time Per Output Token), and will automatically recommend (1) the *energy-optimal* configuration that meets the latency constraints, and (2) the expected amount of energy savings. Due to the generalizability of our measurements (Section 2.1), these recommendations inform the optimization of a wide range of systems.

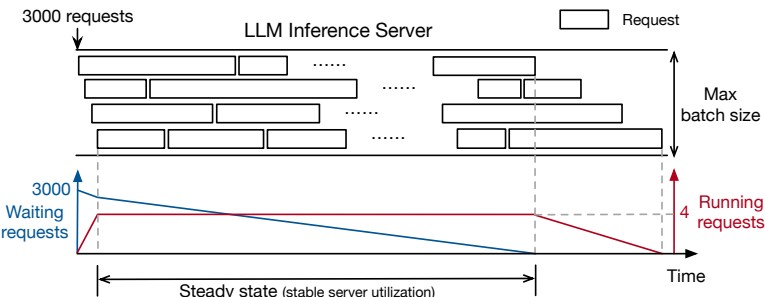

Figure 2: LLM inference server and per-request energy accounting. The steady state is defined as the period when batch size is saturated at the server's maximum configured batch size, and measurements during the steady state represent that of a serving system during long-term deployment.

# 3 The ML.ENERGY Benchmark

The ML.ENERGY Benchmark is a comprehensive tool for measuring and optimizing the inference energy consumption of generative AI models, built upon our core design principles (Section 2). Here, we describe the overall flow of the ML.ENERGY Benchmark (Section 3.1), which includes service-aware energy measurement and accounting (Section 3.2) and automated optimization recommendations (Section 3.3). Finally, we describe extension points of the ML.ENERGY Benchmark that allows users to easily benchmark their customized application scenarios (Section 3.4).

## 3.1 Benchmark Flow

Figure 1 provides an overview of the usage flow of the ML.ENERGY Benchmark. ❶ First, the generative model to benchmark and the request dataset (set of inputs) to use are selected, alongside with the set of configurations to sweep (e.g., GPU model, parallelism configuration, maximum batch size). ❷ Then the ML.ENERGY Benchmark runs configurations independently on designated hardware, and measures the time and energy consumption of each configuration using Zeus [2], a library that provides programmatic energy measurement (Section 3.2). ❸ After benchmarking is complete, users can specify a latency target based on their application requirements. ❹ Given that, the ML.ENERGY Benchmark constructs the time–energy Pareto frontier, and recommends the energy-optimal configuration while satisfying the latency target (Section 3.3).

## 3.2 Energy Measurement and Service-Aware Energy Accounting

Our goal is to provide per-request energy measurements (Section 2.3) that are representative of real-world deployments (Section 2.2). However, a realistic serving system batches together the generation of multiple requests (e.g., iteration-level batching [82] for LLM text generation), making the energy consumption of a single request dependent on all other requests being processed at the same time. Therefore, we implement measurement and energy accounting methods that capture the batching behavior of different types of models.

**Diffusion models.** We begin with the relatively more straightforward case of diffusion models, which are used for text-to-image, text-to-video, and image-to-video generation. Diffusion models are typically batched as a whole, meaning that the energy consumption of a single request is:

$$\text{Energy}_{\text{request}} = \frac{\text{Energy}_{\text{batch}}}{B} \tag{1}$$

where the batch consists of $B$ image or video generation requests.

**LLM text generation.** Request-level energy accounting is less straightforward for LLM inference, because iteration-level batching [82] is an essential optimization in any realistic, production-grade LLM serving system [39]. Figure 2 shows how requests are served by a serving system implementing iteration-level batching and how the ML.ENERGY Benchmark performs energy accounting. Because

the beginning and end of each request are often not aligned with each other, finding each request's individual energy consumption is non-trivial. For this, we first submit all requests in the request dataset, and as the system runs, identify the *steady state* as the time period where the batch size is saturated at the server's maximum configured batch size. This steady state is designed to closely approximate the state of a serving system when it is well-utilized during long-term deployment. Particularly, when the system is ramping up initially with a full queue or ramping down at the end with an empty queue, the server runs with a smaller batch size and does not exhibit the same energy amortization benefits as the steady state. With this, we can derive the average per-request energy consumption with:

$$\text{Energy}_{\text{request}} = \frac{\text{Energy}_{\text{steady}}}{\text{Tokens}_{\text{steady}}} \times \frac{1}{N} \sum_i \text{Tokens}_{\text{request},i}. \tag{2}$$

In essence, we compute the average energy consumption per token during the steady state and multiply it by the average number of output tokens to derive the average per-request energy consumption. Individual requests' energy consumption can also be computed by multiplying the average energy per token during the steady state by the number of output tokens for each request.

As we will see in Section 4, batch size is a critical configuration that significantly affects both generation time and energy consumption. By sweeping the batch size configuration, the ML.ENERGY benchmark can capture varying levels of system utilization and collect various operation points with different time and energy consumption.

### 3.3 Automated Optimization Recommendation

Our goal is to provide actionable insights beyond just energy measurements (Section 2.4) by recommending energy-optimal configurations for a given model and task. Central to the optimization recommendation is the construction of the *Pareto frontier* of energy vs. time, which is a collection of configurations where there are no other configurations that lead to both lower energy and lower time. Then, the energy-optimal configuration is selected based on user-specified latency constraints.

Latency constraints inherently depend on the user's or application's needs. For example, for image generation with Diffusion models, computation results are useful only when the full image is generated, so latency constraints would be specified in terms of the time to generate the whole image. On the other hand, for LLM text generation for chat, output tokens are *streamed* to users (either in written text or synthesized speech) as they are generated. As such, for user-facing conversational AI services, as long as the average time per output token is at least as fast as the users' reading or listening speed, user experience will not be affected [44]. However, for LLM text generation for coding, where code is likely only useful when it is fully generated, latency constraints would be specified in terms of the time to generate the whole snippet, similar to the case of image generation. Given the latency constraints, the time–energy Pareto frontier is used to suggest the minimum-energy configuration that satisfies the latency constraint.

### 3.4 Extending the Benchmark

The ML.ENERGY Benchmark is designed to be easily extensible, allowing users to benchmark their own models or customized application scenarios.

**Model.** The ML.ENERGY Benchmark already supports various popular architectures like Llama [73], LLaVA [43], Stable Diffusion [25], and Stable Video Diffusion [14] (See Appendix A for a full list). Models that are fine-tuned based on already-supported models work as is. Models with different architectures should also work as is as long as they are supported by the underlying runtime, like vLLM [39], which supports arbitrary LLMs provided by Hugging Face Transformers.

**Request dataset.** For each task (e.g., LLM text generation for chat), the ML.ENERGY Benchmark provides a default request dataset that contains a set of inputs representative of real-world usage (See Appendix A for a full list). Users can also provide their own request dataset, which can be used to invoke the runtime and measure energy consumption.

**Configuration space.** The ML.ENERGY Benchmark provides a default set of configurations specific to tasks. For instance, for LLM text generation, it supports maximum batch sizes and

parallelism configuration (e.g., tensor and pipeline parallelism). For diffusion models, it supports not only batch size, but also changing the number of denoising steps, as it has a non-trivial impact on time, energy, and output quality. Users can customize the range of values swept for each configuration, and also provide new configurations (e.g., GPU power limit [1, 81]) as long as they implement the corresponding configuration interface in the top-level routine. More configuration dimensions and finer grained sweeps will lead to longer benchmarking time, but will also push the Pareto frontier towards the lower left corner of the time–energy space, leading to the discovery of more energy-efficient configurations.

**Hardware.** As long as the runtime used by the ML.ENERGY Benchmark (e.g., vLLM) is capable of running on the target hardware and Zeus [2] can measure energy consumption on the target hardware (e.g., NVIDIA/AMD GPUs, Intel/AMD CPUs, Apple Silicon, NVIDIA Jetson platforms), the ML.ENERGY Benchmark can run on the target hardware as is.

**Metrics.** Energy is a fundamental physical quantity that can be used to derive other useful metrics, though these derived metrics are *not* automatically computed by default as they require context-specific information. Below, we describe how these metrics might be computed based on the benchmark's outputs.

- **Average power draw (Watts)**: Average power draw over the steady state can be calculated by dividing total energy consumption during the steady state by the duration of the steady state.
- **Throughput per Watt**: Work throughput, e.g., request or token generation throughput, divided by average power draw can describe how much *service capacity* can be extracted from the system given a power budget, which is a critical quantity for datacenter power planning [38].
- **Monetary cost ($)**: The electricity cost of compute can be calculated by integrating over time the multiplication of energy consumption and the electricity price in the region and time instance. If there is a specific region and time frame the service is expected to run, choosing that electricity price can simulate the operational electricity cost of deployment. Electricity prices can be obtained from sources like OpenEI.[3] Calculating the electricity cost from energy is supported by Zeus [2], the measurement library of choice for the benchmark.
- **Operational carbon emissions ($gCO_2e$)**: This quantity *estimates* the greenhouse gas emissions associated with the electricity consumed. It can be calculated by multiplying energy consumption by the carbon intensity ($gCO_2e$/kWh) of the particular region and time frame in which the benchmark was run. Carbon intensity data can be obtained from sources like ElectricityMaps.[4] This is also supported by Zeus [2], the energy measurement library employed by the benchmark.

## 4 Results Highlight

In this section, we highlight notable results from the ML.ENERGY Benchmark; the full set of results is available on the ML.ENERGY Leaderboard.[5] The early 2025 iteration of the benchmark and leaderboard presents energy measurements across 40 models and 6 tasks (See Appendix A for a full list). We ran the benchmark on NVIDIA A100 (40 GB) and H100 (80 GB) GPUs, each using AWS p4d.24xlarge and p5.48xlarge instances, respectively, and used vLLM [39] and Diffusers [77] as the inference runtime. In the following, we first present energy measurement results and discuss implications (Section 4.1), and then provide deeper understanding by showing how model architecture choices affect their energy consumption (Section 4.2). Then, we present the energy savings opportunities from our automated optimization recommendations (Section 4.3).

### 4.1 Energy Measurements

**Significant variation in energy consumption.** The solid bars in Figure 3 (A100 GPUs in Figure 3a and H100 in Figure 3b) show the per-request energy consumption of various generative AI models across different tasks. First, energy consumption varies widely across models. In particular, Diffusion models generally consume energy that is on par with larger LLMs (e.g., Mistral Large (123B)). This

---

[3]https://openei.org/wiki/Utility_Rate_Database
[4]https://electricitymaps.com/
[5]https://ml.energy/leaderboard

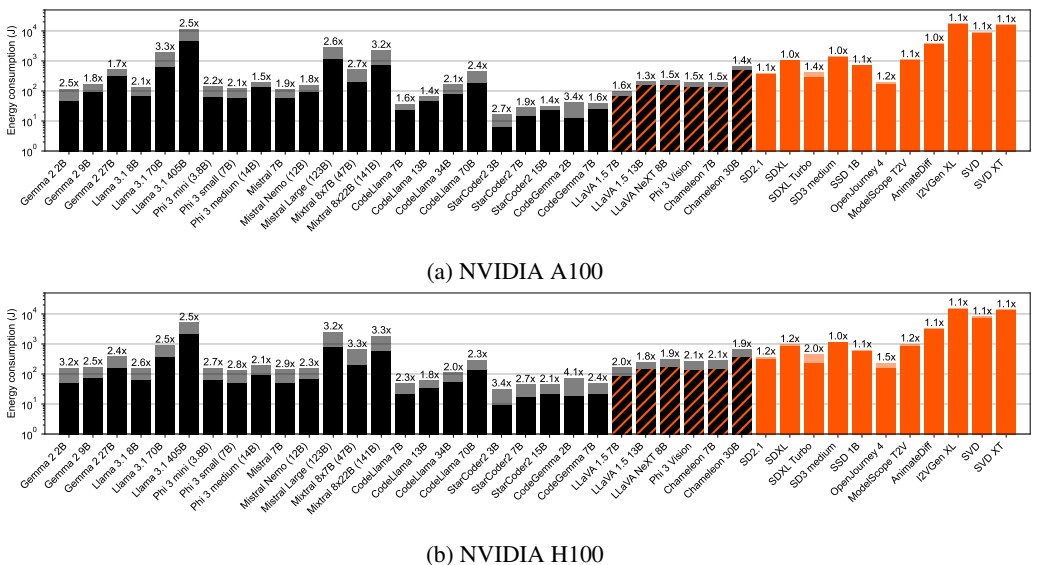

(a) NVIDIA A100

(b) NVIDIA H100

Figure 3: Per-request energy consumption across various generative AI models. Black and orange represents text and vision modalities, respectively. Solid bars are energy measurements, whereas dimmed bars behind each solid bar are estimations based on the GPU's TDP, with numbers showing the ratio of overestimation. Note the log scale Y-axis.

| Model | TP | Max batch size | | | | |
|---|---|---|---|---|---|---|
| | | 4 | 8 | 16 | 32 | 64 |
| DeepSeek distilled Qwen 3 8B [23, 80] | 1 | 9713.7 | 6010.1 | 4314.9 | 3340.8 | 2770.8 |
| Phi 4 reasoning plus 15B [3] | 1 | 19974.4 | 12389.6 | 9347.3 | 7634.9 | 7595.4 |
| Qwen 3 32B [80] | 2 | 26419.7 | 15168.3 | 9140.5 | 6165.5 | 4520.6 |
| Qwen 3 235B-A22B thinking [80] | 8 | 122523.1 | 86491.5 | 56720.4 | 40275.5 | 33096.4 |

Table 1: Energy per generation of reasoning models on GPQA [64] and NVIDIA H100 GPUs. TP is the tensor parallelism degree, which is also equal to the number of GPUs used.

is mainly because Diffusion models (1) draw higher power in general (more in Section 4.2) and (2) cannot perform as many concurrent generations compared to LLMs due to their long latency in real services, preventing them from amortizing energy consumption across many generations.

**Importance of measuring.** The dimmed bars behind each solid bar in Figure 3 show the estimated energy consumption based on the GPU's Thermal Design Power (TDP) instead of measuring the real GPU power consumption, which is a common practice [8, 9, 28, 40, 47, 74]. Estimations using TDP are nearly always an overestimation since it is rare for a GPU – or any computing device – to draw its maximum power at every moment in time. In fact, such an estimation can lead to a worst-case overestimation of energy consumption by a factor of 4.1 (CodeGemma 2B on H100 GPUs). Inaccuracies may be overlooked when they influence downstream decisions and projections, leading to misleading conclusions. Accurate measurements that reflect production environments are crucial.

## 4.2 Energy Implications of ML Design Decisions

ML decisions reflected in model architectures and trained models impact energy consumption. For the interest of space, we defer systems implications on energy consumption to Appendix B.

**LLM response verbosity and energy.** In Figure 3, we can see that energy consumption varies even among LLMs of similar sizes. This is because different LLMs generate responses of different *length* even when given the same prompt. Such differences in *verbosity* can be non-trivial; for instance, Mistral Large's responses were on average 36% longer than that of Mixtral 8×7B. As the number of

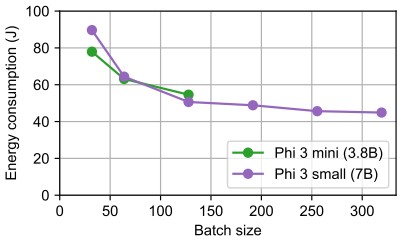
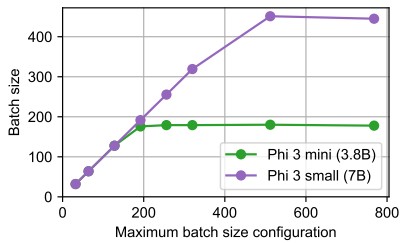

(a) Energy vs. Batch Size     (b) Batch Size vs. Max Batch Size config

Figure 4: Phi-3 Mini and Small [26] benchmarked with the chat task on one NVIDIA A100 GPU.

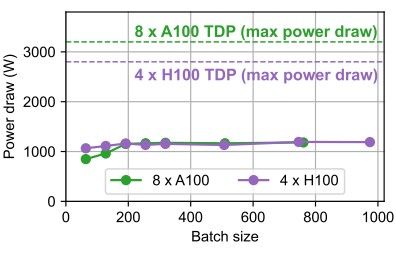
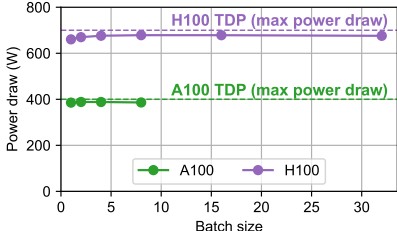

(a) Llama 3.1 70B [73]     (b) Stable Diffusion 3 Medium [25]

Figure 5: Power consumption of Llama 3.1 70B and Stable Diffusion 3 Medium models.

output tokens equals the number of forward passes through the model, longer responses leads to a proportional increase in energy consumption. As humans are known to prefer longer responses [85], this potentially introduces a trade-off between energy consumption and user satisfaction.

This is even more pronounced for reasoning models, which produce significantly more output tokens. Table 1 shows energy measurements for reasoning models on the GPQA dataset. Reasoning models produce one to two orders of magnitude more output tokens per request compared to standard chat models, significantly increasing energy consumption per generation. Additionally, due to their long output lengths, servers cannot run as large a batch size, preventing them from amortizing energy across more requests. This leads to higher energy per token as well, further increasing energy consumption. As long horizon reasoning and task decomposition become more common in real-world LLM-based applications, we expect this trend to continue.

**Memory consumption of operations and energy amortization.** Generally, models with more parameters consume more energy, but this is not always the case. Figure 4 highlights the case of Phi-3 Mini (3.8B) and Small (7B) [26]. Even though Small has nearly twice the parameters, the left plot shows that the larger Small model can consume less energy than Mini as batch size grows. This happens because Mini uses Multi-Head Attention (MHA) [76], whereas Small uses Grouped Query Attention (GQA) [10]. Due to this, Mini's KV cache uses $3\times$ more memory than Small, which prevents it from scaling to larger batch sizes and amortizing energy consumption across more generations.

**Compute-intensity of operations and power draw.** Figure 5 shows the power consumption of Llama 3.1 70B [73] and Stable Diffusion 3 Medium [25] on A100 and H100 GPUs. It can be seen that the LLM's power consumption is much lower than what the GPUs can draw at maximum, whereas the Diffusion model's power consumption is close to the maximum. This is because LLM decoding is characterized by *low compute-intensity*, meaning that the number of arithmetic operations (e.g., multiplication and addition) per byte of memory loaded is low [37, 58]. This leads to the GPU's computation throughput being bottlenecked by VRAM bandwidth and results in the GPU's computation units being underutilized, leading to low power draw. Appendix C dives deeper into power consumption with measurements for all models and GPU power breakdowns over time.

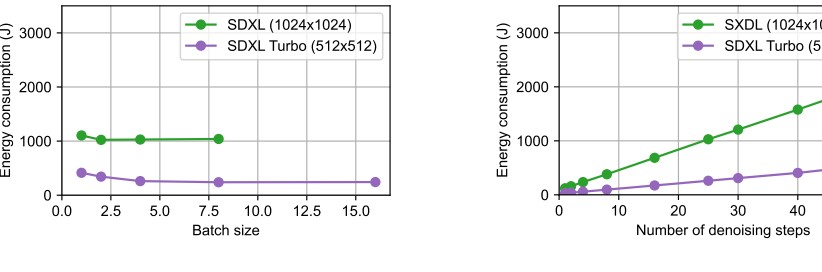

(a) Different Resolutions            (b) Varying Denoising Steps

Figure 6: Energy consumption of SDXL [61] and SDXL Turbo [7] on one NVIDIA A100 GPU.

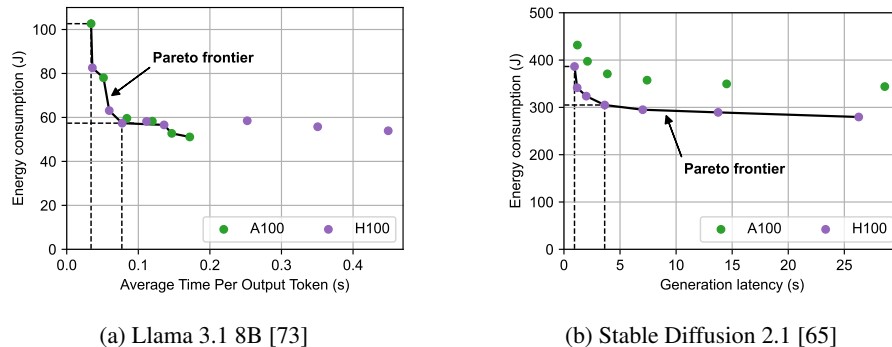

(a) Llama 3.1 8B [73]            (b) Stable Diffusion 2.1 [65]

Figure 7: Time–energy Pareto frontiers constructed by the ML.ENERGY Benchmark.

**Inference-time parameters and energy.** Figure 6 shows the energy consumption of Stable Diffusion XL (SDXL) [61] and SDXL Turbo [7]. On the left, while SDXL and SDXL Turbo have identical model sizes and architectures, their energy consumption is significantly different. This is because SDXL Turbo is tuned to generate smaller resolution images (512×512) than SDXL (1024×1024), which leads to different latent sizes and amounts of computation. On the right, it can be seen that the number of denoising steps linearly increases energy consumption, as one denoising step requires one forward pass through the model. While simple in isolation, these inference-time parameters lead to non-trivial design tradeoffs at the application-level. For instance, increasing the number of denoising steps may improve final image quality, but beyond some point, it may be virtually indistinguishable to human users. Also, generating images in lower resolution and then upscaling them with a separate super-resolution model (e.g., DAT [19]) may consume less energy end-to-end.

### 4.3 Automated Energy Optimization Recommendation

Figure 7 shows the time–energy Pareto frontier constructed by the ML.ENERGY Benchmark measurement results for Llama 3.1 8B and Stable Diffusion 2.1. In general, the Pareto frontier is convex, meaning that by sacrificing some latency, one can achieve significant energy savings.

Conversational AI services like LLM-based chatbots achieve interactivity by streaming tokens to users either in written text or synthesized speech, making Time Per Output Token (TPOT) an important performance metric that impacts user experience [44]. In this context, a chatbot provider can target an average TPOT of 100 ms (equivalent to 10 tokens per second or about 7.5 words per second [55]), which is sufficient for most reading or listening speeds. This will land on the Pareto frontier at the point where average TPOT is 77 ms, reducing energy consumption per generation by 44% compared to the configuration that simply minimizes latency.

Here, we note that for Llama 3.1 8B [73], the Pareto frontier is a mixture of configurations from both A100 and H100 GPUs. This is because LLM decoding does not fully exert the GPU's compute units and are rather bound by memory, so going from A100 to H100 GPUs neither provides significantly higher performance nor significantly increases power draw (See Appendix C for details). These two – power and time – multiplied, energy consumption is comparable across the two GPUs.

On the other hand, for Stable Diffusion 2.1 [65], the Pareto frontier is dominated by configurations on the H100 GPU. Diffusion models consume power close to the GPU's TDP (See Appendix C for details), which increases power draw significantly when going from A100 to H100. However, since computation latency was reduced even more, configurations on H100 Pareto-dominate those on A100. If an application has a generation latency target of, for instance, 5 seconds, the energy-optimal configuration will lie on the Pareto frontier where latency is 3.63 seconds, which is 21% less energy than the configuration that minimizes latency.

## 5 Related Work

**ML energy measurement.** The Hugging Face LLM-Perf leaderboard [33] is specific to LLMs and reports the *per-token* energy consumption of LLM text generation, which fails to capture the verbosity and task-specific output token length distribution difference of LLMs (Section 2.3). MLPerf Power [75] provides measurements for ML training and inference, but crucially, requires direct access to the system under test to physically install the power analyzer, which significantly limits who can run the benchmarks (Section 2.1). Furthermore, it benchmarks at most a few model architectures for each task (sometimes only one), failing to provide insights on how ML design choices impact energy consumption. The Hugging Face AI Energy Score leaderboard [27] provides measurement data for broader AI tasks. However, it fixes the inference batch size to 1 for all models, failing to reflect how services are deployed in the real world and thus their energy consumption (Section 2.2). Google disclosed the median energy consumption of their AI service [24]. It provides a comprehensive scope of measurement, even including the energy consumption of idle machines provisioned for stable service operation. However, measurements and reports are based on internal Google systems, workloads, hardware (TPUs), and model (Gemini) that are not publicly available, limiting the generalizability and reproducibility of the results (Section 2.1). The ML.ENERGY Benchmark is the first inference energy benchmark for modern generative AI models, and empowers users to not only measure but also optimize the energy consumption of their models. See Appendix D for more details.

**ML energy optimization.** The ML.ENERGY Benchmark provides automated energy optimization recommendations based on energy measurements (Section 3.3). There are several other efforts that also provided automated energy optimizations – while preserving mathematical equivalence and/or model quality – for ML training and inference. Zeus [81], EnvPipe [20], and Perseus [21] optimizes the energy consumption of ML training by adjusting GPU-level and training job-level configurations, either statically after profiling or dynamically during training. $\mu$-Serve [63] and DynamoLLM [68] are also similar, but optimize energy consumption for ML inference clusters. Optimization recommendations by the ML.ENERGY Benchmark are complementary to the techniques proposed by these works. Further, our results support the need for automated *cross-layer* energy optimizations that span all model, software, and hardware layers [22], as opposed to efforts siloed within a single layer.

## 6 Conclusion

In this work, we described the ML.ENERGY Benchmark, a comprehensive energy benchmark for generative AI models that not only provides realistic energy measurements, but also automatically suggests energy-optimal configurations based on user- and app-specific performance constraints. Measurement results show that energy consumption is a metric that is impacted by design choices across the whole AI stack, including application, model, software, and hardware, demonstrating the importance of automated *cross-layer* energy optimizations instead of siloed optimizations within a single layer. We are confident that the ML.ENERGY Benchmark will democratize the art of measuring, understanding, and optimizing ML energy consumption for the community.

## Acknowledgments and Disclosure of Funding

We would like to thank Yunseok Jang and SymbioticLab members for helpful comments and suggestions on the paper. This work and its authors were in part supported by NSF grants CNS-2104243, CNS-2106184, and CNS-2450085, grants from VMware, the Mozilla Foundation, Cisco, Ford, and GitHub, and gifts from Salesforce and Google. Jae-Won Chung is additionally supported by the Kwanjeong Educational Foundation.

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

Table 2: Model type, task, and default request dataset used in the ML.ENERGY Benchmark.

| Model architecture | Task | Request dataset |
|---|---|---|
| Large Language Model | Chat | ShareGPT [72] |
| | Code | EvalPlus [45] |
| Vision Language Model | Visual chat | LLaVA instruction dataset [43] |
| | Text-to-image | PartiPrompts [83] |
| Diffusion Model | Text-to-video | Captions in ShareGPT4Video [18] |
| | Image-to-video | Captions and first frames in ShareGPT4Video [18] |

Table 3: Model architectures supported by the ML.ENERGY Benchmark for each task.

| Task | Model Architectures |
|---|---|
| Chat | Gemma 2 2B/9B/27B [71], Llama 3.1 8B/70B/405B [73], Phi 3 Mini/Small/Medium [26], Mistral 7B/Nemo/Large [34], Mixtral 8x7B/8x22B [35] |
| Code | CodeLlama 7B/13B/34B/70B [66], StarCoder 2 3B/7B/15B [46], CodeGemma 2B/7B [70] |
| Visual chat | LLaVA 1.5 7B/13B [41], LLaVA NeXT 8B [42], Phi 3 Vision [26], Chameleon 7B/30B [69] |
| Text-to-image | Stable Diffusion 2.1/XL/XL Turbo/3 Medium [7, 25, 61, 65], OpenJourney 4 [62], SSD 1B [30] |
| Text-to-video | ModelScope T2V [78], AnimateDiff [29] |
| Image-to-video | I2VGen XL [84], Stable Video Diffusion and Stable Video Diffusion XT [14] |

# A    Tasks, Model Architectures, and Default Request Datasets

Tables 2 and 3 list the model architectures and tasks supported by current iteration of the ML.ENERGY Benchmark, along with the default request datasets for each task. We note that models that were fine-tuned based on the supported models are also supported as is, and the benchmark is designed to be extensible (Section 3.4).

The ML.ENERGY Benchmark cannot avoid being outdated given the rapid pace of development in the generative AI field. As such, we have been updating the benchmark (and the accompanying Leaderboard) with new tasks, models, datasets, hardware, runtimes, and more, and we intend to continue doing so as long as resources allow.

# B    Energy Implication of System Parameters

This section discusses the energy implication of different system-level configurations. System-level configurations are those that do not change *what* is computed but rather *how* it is computed by the underlying software system.

## B.1    Request Preemption Mechanism

Even with the model and inference parameters fixed, the software system used to serve inference requests, which determines how model computations are executed on a given hardware, significantly impacts energy consumption. As a concrete example, we will examine the effect of "preemption mechanism," a configuration parameter for LLM inference servers. When a server is overloaded with more requests than its capacity, it needs to temporarily remove (or, preempt) some requests from the system and then later bring them back (or, restore). For LLM inference, there are two widely-used mechanisms for preemption: Recomputation and Swapping [39]. Recomputation simply drops all temporary request data or state on preemption and recomputes everything from scratch on restoration. On the other hand, Swapping moves the request state to the CPU's memory, and then returns it to the GPU on restoration. The best preemption mechanism depends on the computing hardware and software configuration and the LLM being served.

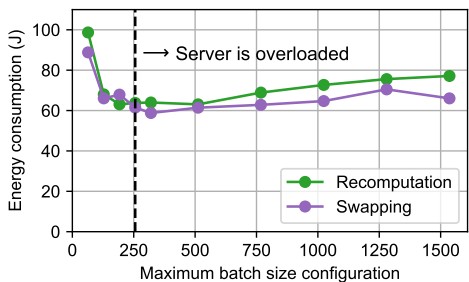

Figure 8: Energy consumption per generation while varying the maximum batch size for Mistral Nemo (12B). The LLM inference server's preemption mechanism is compared.

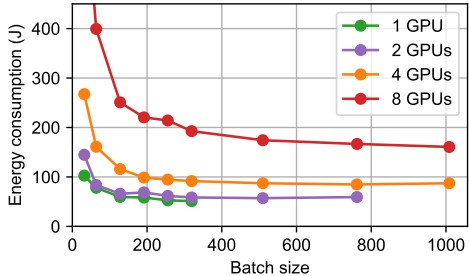

Figure 9: Energy consumption per generation while varying batch size for Llama 3.1 8B. The number of NVIDIA A100 GPUs used to run the same model is scaled up.

Figure 8, we compare the energy consumption per generation of the two preemption mechanisms with the Mistral Nemo (12B) model by intentionally overloading the server with a high maximum batch size configuration and causing preemption. It can be seen that when the server is overloaded, Swapping consistently consumes less energy. This is because Recomputation performs extra computation when restoring requests whereas Swapping copies data without running computation, and the energy consumption of computation is larger than memory operations (this will be further examined in the next section). Furthermore, as the server gets more and more overloaded, energy consumption generally increases. This is because with higher overload, more preemptions – and thus more recomputation or data movement – occur. Since preemptions do not directly contribute to the completion of the request, the extra energy consumption from preemptions increases the average energy consumption of completing each request.

### B.2  Tensor Parallelism Scaling

We investigate the impact of communication overhead to energy consumption. This is important as modern large models frequently do not fit within the memory capacity of a single GPU. This requires multiple GPUs to execute inference for a single model, and GPUs must constantly communicate with each other to do so [67].

In order to ablate the effect of communication, we employ the same Llama 3.1 8B model and vary the number of GPUs used (Figure 9). Because the amount of computation executed is the same regardless of the number of GPUs, energy consumption should ideally be constant. Indeed, energy consumption barely changes when scaling from one GPU (no communication) to two, but when scaling further, energy consumption significantly increases. This is because, while the amount of computation decreases for each GPU, additional communication time between the GPUs offsets the reduction in computation time. Since communication time increases with the number of GPUs, using too many GPUs can lead to slowdowns in executing the same amount of computation and increase energy consumption.

| Model and deployment | Request dataset | | |
|---|---|---|---|
| | Input mean 512 Output mean 512 | Input mean 512 Output mean 4096 | Input mean 4096 Output mean 512 |
| Llama 3.1 8B (TP=1, 1P3D) | 37.71, 77.2% | 665.77, 98.7% | 208.34, 67.2% |
| Llama 3.1 8B (TP=1, 2P2D) | 36.22, 76.7% | 706.27, 98.8% | 151.75, 55.2% |
| Llama 3.1 8B (TP=1, 3P1D) | 37.26, 77.0% | 748.45, 98.9% | 158.85, 56.0% |
| Llama 3.1 70B (TP=4, 1P1D) | 276.93, 64.8% | 907.60, 89.2% | 1492.59, 50.0% |

Table 4: Energy per generation (Joules) and the percentage of decode energy consumption with PD disaggregation. Following recent trace analysis [79], we sampled input lengths from a Pareto distribution with alpha 2.5, and output lengths from an Exponential distribution, each with mean specified in the table. TP means tensor parallelism degree, and xPyD means it was deployed with $x$ prefill instances and $y$ decode instances, each with TP-many GPUs.

| Max batch size (sequences) | Max batched tokens | | | | | | | | |
|---|---|---|---|---|---|---|---|---|---|
| | 32 | 64 | 128 | 256 | 512 | 1024 | 2048 | 4096 | 8192 |
| 32 | 559.66 | 374.63 | 269.29 | 205.54 | 188.80 | 191.61 | 195.59 | 191.88 | 194.52 |
| 64 | | 362.49 | 266.98 | 200.43 | 168.27 | 165.52 | 170.17 | 168.78 | 169.58 |
| 128 | | | 264.18 | 194.59 | 164.75 | 154.64 | 155.59 | 156.54 | 156.93 |
| 256 | | | | 194.39 | 161.87 | 153.97 | 155.11 | 157.13 | 159.25 |
| 512 | | | | | 159.57 | 151.50 | 154.52 | 156.77 | 154.95 |
| 1024 | | | | | | 152.67 | 156.26 | 157.98 | 163.08 |

Table 5: Energy per generation (Joules) of Llama 3.1 8B on a synthetic long context request dataset running on H100 GPUs. Following recent trace analysis [79], we sampled input lengths from a Pareto distribution with mean 4,096 and alpha 2.5, and output lengths from an Exponential distribution with mean 512. Note that vLLM does not allow the max number of batched tokens to be smaller than the max batch size, which is why the lower left triangle of the table is empty.

From this scaling experiment, we can observe that the energy impact of communication overhead can be large. This impact will be even more pronounced in hardware environments without sufficient or state-of-the-art networking infrastructure, which is common in real world settings due to its cost [36].

### B.3 Prefill–Decode Disaggregation

Prefill–decode (PD) disaggregation is a rising production deployment setting where prefill and decode phases are run on separate GPUs [59, 86]. This allows for independent scaling and optimization of prefill and decode phases based on workload characteristics, and leads to better latency deadline attainment. Table 4 shows energy measurements for different PD disaggregation configurations, where "xPyD" denotes $x$ prefill instances and $y$ decode instances.

Overall, decode consumes the majority of energy, with some amount shifting to prefill when input length is long. In our setup, PD disaggregation configurations did not have a large impact on absolute energy consumption or the energy split as long as the throughput of prefill and decode instances are reasonably balanced.

### B.4 Chunked Prefill

Chunked prefill is a technique where long input prompts are split into chunks and processed alongside decode iterations, improving GPU utilization and reducing the interference between long prefills and decode iterations [5]. For chunked prefill, the max number of batched tokens is a key parameter that controls the chunk size. Table 5 shows the impact of this parameter on energy consumption.

Table 5 shows that the more sequences or tokens you batch, the better the energy amortization you get and energy per generation decreases, and after a certain point, returns diminish.

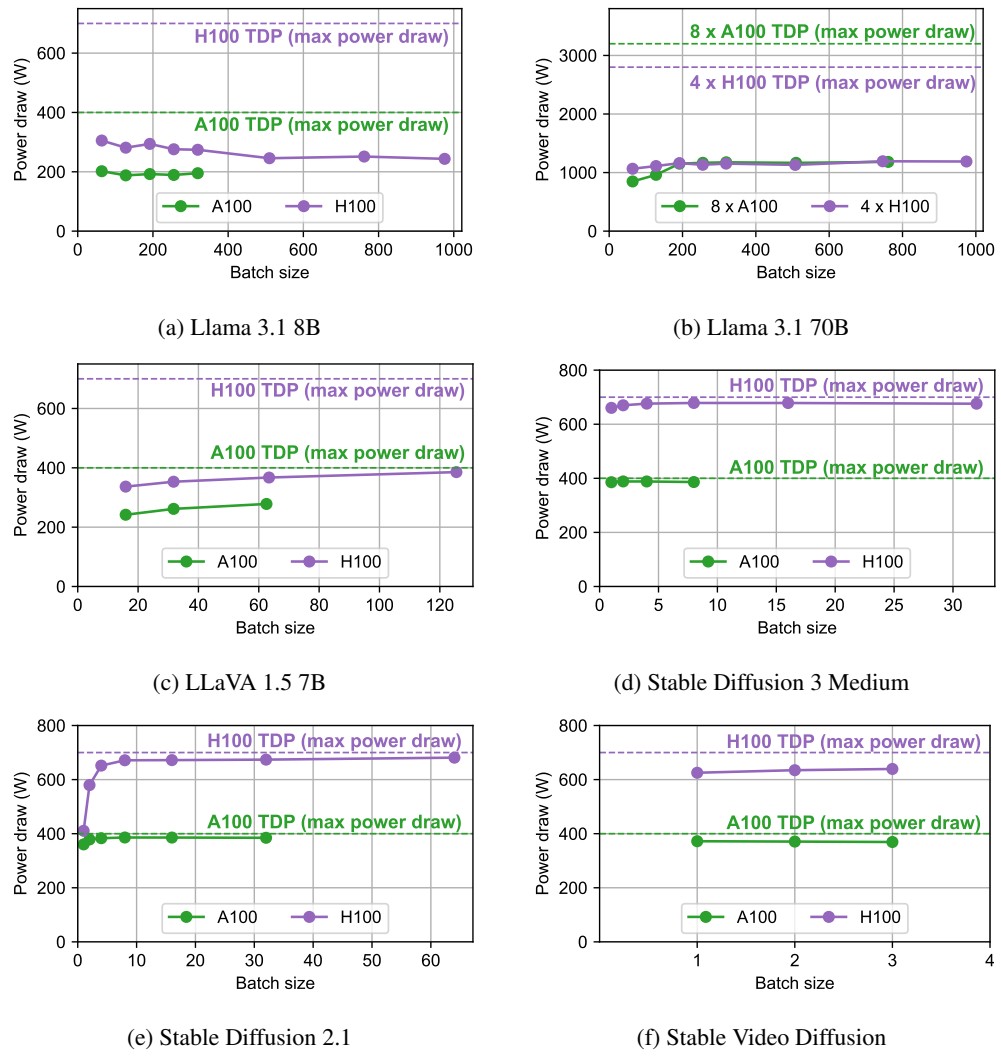

| | |
|---|---|
| (a) Llama 3.1 8B | (b) Llama 3.1 70B |
| (c) LLaVA 1.5 7B | (d) Stable Diffusion 3 Medium |
| (e) Stable Diffusion 2.1 | (f) Stable Video Diffusion |

Figure 10: Power consumption of various models on A100 and H100 GPUs.

## C   Power Consumption Analysis

Figure 10 shows the power consumption of various models on A100 and H100 GPUs. Figure 11 further shows the ratio of a model's power consumption to the maximum GPU power draw across all models. Generally, LLMs and VLMs consume significantly less power than the GPU's TDP because LLM decoding, the dominant operation for LLM serving, is memory-intensive and does not fully utilize the GPU's compute resources. VLMs show slightly higher power consumption than LLMs due to its additional modality encoder, which is compute-intensive. Diffusion models, on the other hand, consume nearly the maximum power of the GPU when batch size is not small. This is because Diffusion models are significantly more compute-intensive compared to LLM decoding.

Figure 12 shows the GPU power draw breakdown over time on one NVIDIA H100 GPU. The GPU's power is measured (1) in whole and (2) only for the VRAM while the ML.ENERGY Benchmark is running. First, for Llama 3.1 8B, the timeline shows the effect of the two phases in LLM text generation: Prefill and Decode. Prefill happens once at the beginning of a request to digest the input prompt, which is then followed by hundreds to thousands of Decode phases, each of which generates one output token. Importantly, Prefill has high compute-intensity (and also high power draw) because it needs to digest the whole input prompt, whereas Decode has low compute-intensity (and low power draw) as it does not entail very much computation. With this, we can first understand the initial spike

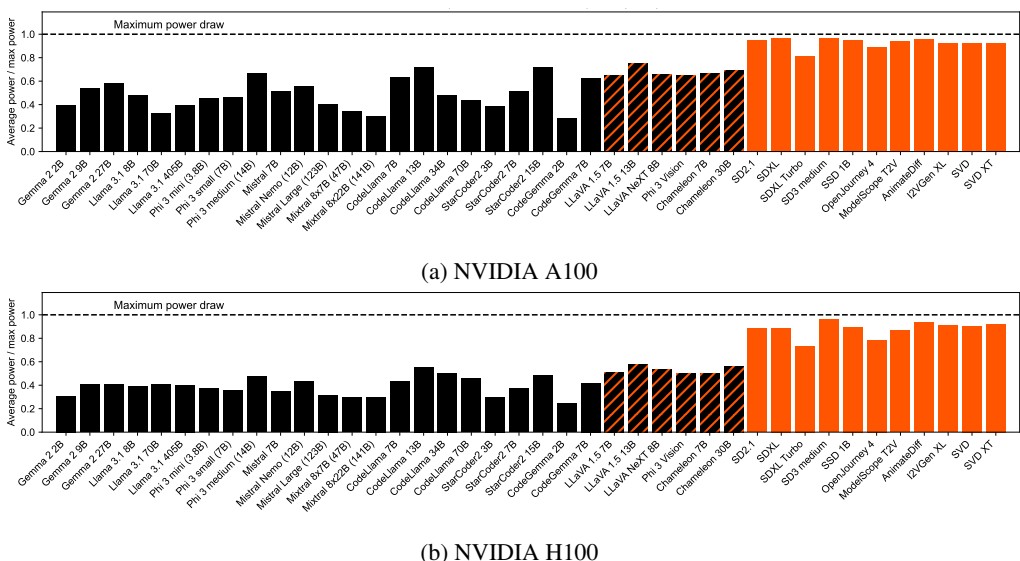

(a) NVIDIA A100

(b) NVIDIA H100

Figure 11: Ratio of power consumption to maximum GPU power draw across various models.

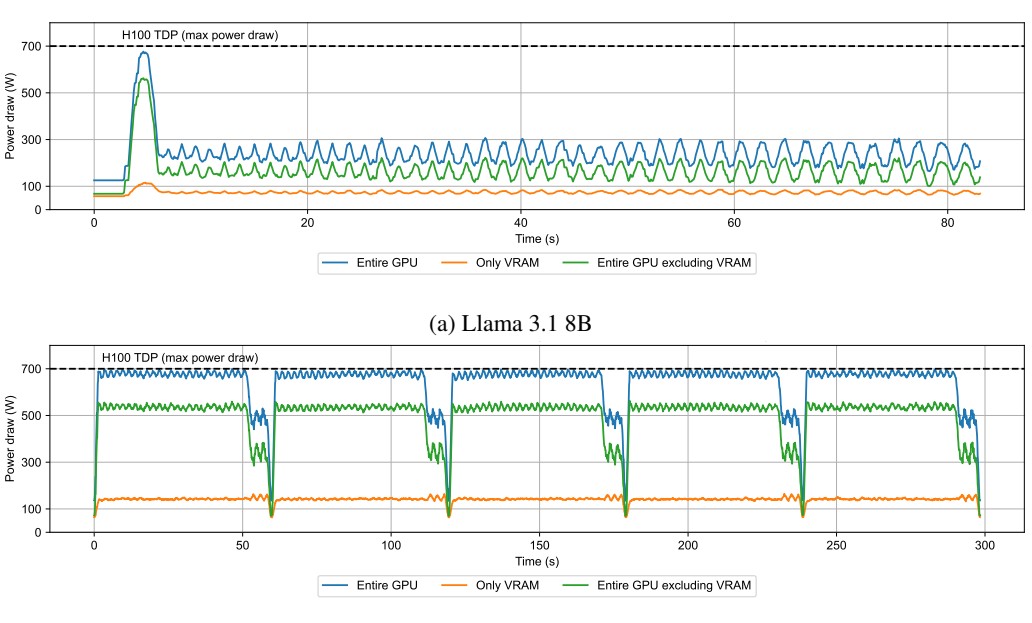

(a) Llama 3.1 8B

(b) Stable Video Diffusion XT

Figure 12: GPU power draw breakdown over time on one NVIDIA H100 GPU. "Entire GPU" and "Only VRAM" (memory) were measured, and the two were subtracted to derive "Entire GPU excluding VRAM."

in power draw – when the benchmark begins, the server begins admitting new requests, creating a short period where numerous Prefills are executed back-to-back, leading to high power draw. After the initial spike, power draw repeats a periodic fluctuation. This is because, before each Prefill or Decode, the server must make numerous control decisions, including determining which requests are now finished and which ones should run next. Since these decisions are executed by the CPU, this creates a periodic time gap where the GPU is not running any computation. This GPU idle time leads to the periodic drop in GPU power draw.

On the other hand, Stable Video Diffusion XT shows a different power draw pattern. Diffusion models generally have three phases: Encode, Denoise, and Decode. The Encode phase digests the

input prompt and passes it to the Denoise phase, which iteratively removes noise from a random vector. Finally, the Decode phase transforms the denoised vector into the final image or video.

From the timeline, especially Denoise and Decode can be clearly distinguished. Denoise is the most compute-intensive and consumes power close to the GPU's TDP. For each batch, there are 25 local peaks that hit the GPU's TDP, each of which corresponds to one denoising step in Denoise. During Decode, power draw generally decreases, with each local power peak corresponding to the two large layers in the decoding module. On the other hand, VRAM power draw increases during Decode because it allocates a large chunk of memory and performs writes in order to generate the final video. Finally, as the final generated video is copied from the GPU's memory to the CPU's, the GPU does not run any computation, resulting in a steep drop in power draw.

From the power breakdown, we can observe that memory operations indeed draw significantly less power compared to computation, and thus computations with low compute-intensity should indeed draw less power. Furthermore, we can observe that the power draw and energy consumption of a specific hardware (GPU in this case) is not a function of just itself and the computations that it runs. Rather, software and hardware components that are integrated in the same system stack impacts how computations are executed on each other, affecting their power draw and energy consumption.

## D   The ML.ENERGY Leaderboard and Benchmark

On July 2023, we launched the ML.ENERGY Leaderboard and Benchmark, the first inference energy leaderboard for modern generative AI models.[6] Our goal was to measure and understand the energy consumption of generative AI models, and we provided a web-based leaderboard to allow everyone to browse the results. The leaderboard started with only LLM chat with tens of different LLMs, but gradually expanded to include more tasks, models, and datasets. Our benchmarking suite to supply data to the leaderboard is what we dub the ML.ENERGY Benchmark. This paper shares our design philosophy and principles we have acquired over time by gradually maintaining and upgrading the ML.ENERGY Benchmark and the Leaderboard, and highlights notable results we have obtained from the early 2025 iteration of the benchmark. Importantly, we plan to continuously update the benchmark and the leaderboard as long as resources allow, and what is presented in this paper is only a snapshot of the current state of the benchmark at the time of writing. We encourage readers to visit the leaderboard website and benchmark repository for the latest results and updates.

## E   Limitations

The ML.ENERGY Benchmark is not without limitations. First, we note that the benchmark is not exhaustive and does not cover all possible tasks, models, and datasets. This is particularly true as time passes and new models and tasks are developed. We are aware of newer open-weight models and worthy tasks that were released after the early 2025 iteration of the benchmark was finalized. However, we cannot add each model or task one by one incrementally as they are released, due to the prohibitive monetary cost of running the benchmark on representative hardware; rather, we collect new advances in a window of time and then mass-update the whole benchmark, accompanied by upgrades in hardware, software, and datasets. Second, the benchmark is not exhaustive in terms of hardware. We currently mainly support flagship NVIDIA GPUs, which arguably dominates the market especially when it comes to real-world generative AI services. Furthermore, we do not have access to all possible hardware configurations, nor do they always provide a way for us to measure energy consumption from software. Regardless, we are working to expand the benchmark to support more hardware configurations.

## F   Broader Impacts

By allowing everyone to accurate measure, understand, and optimize the energy consumption of generative AI models, we believe the ML.ENERGY Benchmark can enhance the understanding of energy consumption of generative AI in the research community and the industry, and ultimately fuel works that optimize energy consumption. Furthermore, energy is essentially throughput per watt,

---

[6]https://github.com/ml-energy/leaderboard/releases/tag/2023-07-06

which is one factor that determines the cost of running generative AI services at the infrastructure level. By optimizing energy consumption, we can reduce the cost of running generative AI services, which can help democratize access to generative AI.

