# OpenReview forum: "The ML.ENERGY Benchmark: Toward Automated Inference Energy Measurement and Optimization"
_NeurIPS.cc/2025/Datasets_and_Benchmarks_Track — NeurIPS 2025 Datasets and Benchmarks Track spotlight_

### Official Review · Reviewer_JV8P · 2025-07-03

**Rating:** 4
**Confidence:** 3

**Summary:**

The authors introduce ML ENERGY, an open-source benchmark and tooling suite that turns raw NVML power traces into a clean joules-per-request metric for deployed generative-AI workloads. They combine a marker-based logging pipeline (which trims warm-up, idle tails and handles dynamic batching) with an automatic Pareto search that sweeps batch size, precision, queue budget and other knobs until it finds the lowest-energy configuration that still satisfies a user-specified latency SLO. The release ships with > 40 open-source text/code/vision-generation models profiled on A100 and H100 GPUs, plus all traces, scripts and a public leaderboard.
Overall I think this benchmark is a solid contribution to the field and provides a tool as well as a leaderboard to demonstrate the energy efficiency of various Gen-AI models.

**Dataset Code Accessibility:**

Yes

**Ethical Considerations:**

No, there are no or only very minor ethics concerns

**Limitations Weaknesses:**

- Hardware scope is narrow. All results use NVML on A100/H100. These two covers the bulk of current cloud inference, but skips AMD, TPU and the growing class of custom inference ASICs. Even a single “how-to add a new backend” example would help jump-start contributions.
- GPU-only energy. Host-side CPU, DRAM, NIC and facility PUE are ignored. The authors argue the GPU dominates joules/request for large LLMs (fair), yet a short node-level validation—say, correlating NVML with wall-plug power on one server—would strengthen the case.
- A potential concern arises from the "steady state" methodology used for LLM energy accounting. The paper defines this state as the period when the server is fully saturated, and explicitly excludes the initial ramp-up and final ramp-down phases from the calculation. While the reasoning—to capture the state of a well-utilized system—is clear, this choice means the resulting "per-request energy" metric represents a best-case, peak efficiency scenario. This is an optimistic measurement that may not fully reflect the total energy cost in real-world deployments where fluctuating demand makes the less-efficient ramp-up and ramp-down periods unavoidable.

**Strengths Contributions:**

- Timely and practically important. As energy consumption becomes an increasingly significant part of the operational cost for large-scale LLM deployments, having a reproducible and standardised way to measure and compare energy use is both necessary and overdue.
- Service-aware methodology. I like that the authors measure per request, not per batch, and that they factor in realistic prompt-length distributions, dynamic batching, and queue-time budgets. The steady-state accounting makes the numbers believable.
- One-click optimisation loop. The built-in Pareto sweep answers the question practitioners actually care about—“what’s the greenest config that still hits my p99 latency target?”—rather than dumping a single batch-size result.
- Open and well documented. Everything (code, Dockerfile, raw CSV traces) is Apache-2.0.

---

> ### Author Rebuttal · Authors · 2025-07-30
>
> We sincerely appreciate your feedback; it will improve not only the current submission, but also all future iterations of the benchmark.
>
> > Hardware scope is narrow. All results use NVML on A100/H100. These two covers the bulk of current cloud inference, but skips AMD, TPU and the growing class of custom inference ASICs. Even a single “how-to add a new backend” example would help jump-start contributions.
>
> We have looked into TPUs and have discussed with Google engineers directly; there is currently no way to measure a TPU’s power/energy consumption. AMD GPUs do have a similar API, but it contains a critical bug that returns wrong values. Our team discovered the problem and filed an issue (GitHub ROCm/AMDSMI repository issue #38), but it has not been resolved for a long time.
>
> In sum, the minimum requirement for a hardware backend to be supported is that it should allow power or energy measurement (only power is fine, as power can be integrated over time to compute energy). Using a hardware backend that is already supported by the runtime (e.g., vLLM) requires no additional work. On the other hand, swapping out the runtime to one that supports a new hardware is also relatively simple. For instance, since all production-ready LLM inference servers support the OpenAI API schema, all the user needs to do in order to upgrade the runtime or use a different one is to extend the benchmark’s logic to spin up the server Docker container. We will definitely add more guides on extending the benchmark.
>
> > GPU-only energy. Host-side CPU, DRAM, NIC and facility PUE are ignored. The authors argue the GPU dominates joules/request for large LLMs (fair), yet a short node-level validation—say, correlating NVML with wall-plug power on one server—would strengthen the case.
>
> We note that works from hyperscalers \[1\] have already indicated that GPUs are allocated 2/3 of the server’s computing equipment power. As researchers, we unfortunately do not own such production-grade hardware (even if we did, it will no longer be production-grade in a very short time span), which prevents us from producing representative experiment numbers in that respect. Moreover, we are yet to find a cloud vendor with flagship GPUs that also provide CPU/DRAM power measurements (via Intel RAPL) or whole-server power measurements.
>
> However, still, GPUs are the primary workhorse of ML compute, whereas all other computing equipments exist to feed the GPU with data and are not a bottleneck or a significant power consumer on a well-designed production system. Furthermore, CPUs run non-ML work like the kernel, system daemons, shells, and more, and energy consumption of such work is inevitably conflated with what is consumed by the ML workload, making measurements misleading and optimization difficult. This is why we believe measuring and optimizing only the GPU’s energy consumption is still very much useful.
>
> \[1\] Patel et al. 2024\. Characterizing Power Management Opportunities for LLMs in the Cloud.
>
> > A potential concern arises from the "steady state" methodology used for LLM energy accounting. The paper defines this state as the period when the server is fully saturated, and explicitly excludes the initial ramp-up and final ramp-down phases from the calculation. While the reasoning—to capture the state of a well-utilized system—is clear, this choice means the resulting "per-request energy" metric represents a best-case, peak efficiency scenario. This is an optimistic measurement that may not fully reflect the total energy cost in real-world deployments where fluctuating demand makes the less-efficient ramp-up and ramp-down periods unavoidable.
>
> This is a great point; we are glad you mentioned it. We can think of two types of load fluctuations: (1) long term fluctuations (hours or days) and (2) short term fluctuations (minutes). We can safely ignore the former, because they will be handled by the production system’s autoscaler; with a diurnal increase in request rate, more server instances will be deployed, and vice versa. On the other hand, short term request load fluctuations over time manifest themselves as changing batch size over time, which is precisely what we intend to capture by varying batch size in our sweep.
>
> Our web leaderboard shows this when the user ticks the “Show more technical details” checkbox and adjusts the target latency slider. Table 1 also shows a quick summary. The general trend is that with larger batch size, energy per generation decreases, but eventually with diminishing returns.
>
> ---
> **Table 1\. Energy per generation (Joules) of Llama 3.1 8B, Llama 3.1 70B, and Stable Diffusion 3 Medium on H100 GPUs across varying max batch sizes.**
>
> | Model \\ Max batch size | 64 | 128 | 256 | 512 | 1024 |
> | :---- | ----: | ----: | ----: | ----: | ----: |
> | Llama 3.1 8B | 82.59 | 63.12 | 58.20 | 58.49 | 53.93 |
> | Llama 3.1 70B | 605.48 | 443.21 | 354.31 | 306.96 | 281.61 |
>
> | Model \\ Batch size | 2 | 4 | 8 | 16 | 32 |
> | :---- | ----: | ----: | ----: | ----: | ----: |
> | Stable Diffusion 3 Medium | 1187.25 | 1141.11 | 1124.41 | 1115.08 | 1111.71 |
> ---
>
> With this, those who would like to simulate load fluctuations and the resulting energy consumption changes can run the benchmark (with potentially more fine-grained batch size ranges), and put together resulting measurements based on their service’s load profile over time.

---

### Official Review · Reviewer_c1Qn · 2025-07-03

**Rating:** 4
**Confidence:** 3

**Summary:**

This paper introduces *ML.ENERGY*, a benchmark designed to measure inference energy consumption in realistic service environments. It features a leaderboard that reports energy usage across 40 model architectures spanning 6 diverse tasks. The benchmark employs production-grade software and hardware, running models on real-world use cases to capture steady-state energy consumption for accurate and consistent measurement. Results highlight that design decisions across the entire AI stack significantly affect energy efficiency, underscoring the importance of cross-layer optimization.

**Additional Feedback:**

N/A

**Dataset Code Accessibility:**

Yes

**Dataset Code Comments:**

**Motivation:**
The benchmark aims to accurately estimate energy consumption for various model–hardware combinations, enabling informed optimization to reduce energy use in real-world settings.

**Reproducibility:**
The paper provides sufficient detail on hardware, software stack, and measurement methodology to support reproducibility. However, due to the rapid evolution of inference software and sensitivity to CUDA kernel implementations, results may vary over time without strict version control.

**Ethical Considerations:**

No, there are no or only very minor ethics concerns

**Final Justification:**

Same as mentioned in my comment, based on the updates provided in the rebuttal, I am inclined to raise my score.

**Limitations Weaknesses:**

While the proposed benchmark closely reflects real-world use cases, its results are highly dependent on the choice of benchmark datasets and the specific software stack used. Rapidly evolving systems—such as vLLM, which is advancing at a rapid pace—can introduce significant inconsistencies between benchmark runs. This suggests the benchmark must be continuously updated to stay relevant, which in turn limits the stability and long-term interpretability of its conclusions. As a result, the energy-saving insights provided may remain coarse-grained and lack fine-grained fidelity.

Moreover, the benchmark results can be sensitive to low-level software implementation details, such as CUDA kernel optimizations. Variations in kernel efficiency, scheduling, and fusion strategies can significantly affect energy consumption, even for the same model and hardware configuration. Without controlling or accounting for these differences, the benchmark may conflate software engineering improvements with hardware or model design choices.

In addition, the study focuses exclusively on GPU hardware, omitting other accelerators like TPUs, which limits the generalizability of its findings. For large language models (LLMs) in particular, the benchmark would benefit from including more representative inference scenarios—such as disaggregated profiling and decoding, and chunked prefill—which are commonly used in production and have distinct energy profiles that merit evaluation.

**Strengths Contributions:**

This paper is well-written and comprehensive. It includes a rich set of figures that effectively illustrate how the benchmarks are conducted and clarify the concept of *steady-state* energy consumption. While prior work has explored energy efficiency across various model–hardware combinations, much of it focuses on inference latency and overlooks the realistic scenario where requests are processed in batches. In contrast, this paper emphasizes commonly deployed software–hardware stacks and evaluates energy efficiency based on steady-state performance under batch inference. Notably, the authors measure actual GPU power consumption rather than relying on Thermal Design Power (TDP), which is known to significantly overestimate real energy usage. This leads to more accurate and representative results. Lastly, the benchmark provides valuable insights into how hardware choices affect the energy consumption of different models, offering practical guidance for energy-efficient system design.

---

> ### Author Rebuttal · Authors · 2025-07-30
>
> We sincerely appreciate your feedback; it will improve not only the current submission, but also all future iterations of the benchmark.
>
> > While the proposed benchmark closely reflects real-world use cases, its results are highly dependent on the choice of benchmark datasets and the specific software stack used. Rapidly evolving systems—such as vLLM, which is advancing at a rapid pace—can introduce significant inconsistencies between benchmark runs. This suggests the benchmark must be continuously updated to stay relevant, which in turn limits the stability and long-term interpretability of its conclusions. As a result, the energy-saving insights provided may remain coarse-grained and lack fine-grained fidelity.
> >
> > Moreover, the benchmark results can be sensitive to low-level software implementation details, such as CUDA kernel optimizations. Variations in kernel efficiency, scheduling, and fusion strategies can significantly affect energy consumption, even for the same model and hardware configuration. Without controlling or accounting for these differences, the benchmark may conflate software engineering improvements with hardware or model design choices.
>
> We agree that this is a rapidly evolving field and that optimizations across the stack impact both performance and energy benchmarking results. We believe this only makes proper benchmarking more important, as making rapid progress without a very good sense of current state and trend is dangerous, and due to the velocity of the field, the benchmark deserves to be run more often.
>
> This is one of the fundamental principles that underpin the design of the benchmark. We wish to capture the **current** state of matters at the time the benchmark is run by freezing the model, software, and hardware stack, so that fair comparisons between tasks and models are made on top of a fixed platform. We also allow new models to be easily included into the benchmark, and allow execution runtimes to be easily swapped out (e.g., since all production-ready LLM inference servers support the OpenAI API schema, all the user needs to do in order to upgrade the runtime or use a different one is to extend the benchmark’s logic to spin up the server Docker container).
>
> Finally, while the field itself is rapidly moving, we would like to note that it is also not without (near) invariants. Autoregressive decoder-only Transformer models are being used for a long time, its arithmetic intensity characteristics (e.g., compute-bound prefill and memory-bound decode) will likely remain similar because it is a mathematical property of that operation, and flagship models from big model builders (e.g., Llama and Gemma) are used widely for multiple months. Therefore, results that are a function of these slowly changing dimensions – for instance, LLM decoding consuming less power over time than prefill, or energy consumption per generation being influenced significantly by LLM verbosity – will have equivalently long shelf lives.
>
> > In addition, the study focuses exclusively on GPU hardware, omitting other accelerators like TPUs, which limits the generalizability of its findings.
>
> We have looked into TPUs and have discussed with Google engineers directly; there is currently no public method to measure a TPU’s power/energy consumption. AMD GPUs do have a similar API, but it contains a critical bug that returns wrong values. Our team discovered the problem and filed an issue (GitHub ROCm/AMDSMI repository issue #38), but it has not been resolved for a long time.
>
> ML energy measurement and optimization is a relatively new field of study, and naturally, it lacks many of the essential tools. Through our work, we hope to convince the broader community of the importance of this field, and to push for efforts to close these gaps.
>
> > For large language models (LLMs) in particular, the benchmark would benefit from including more representative inference scenarios—such as disaggregated profiling and decoding, and chunked prefill—which are commonly used in production and have distinct energy profiles that merit evaluation.
>
> We agree that it is very much worth it to benchmark prefill–decode disaggregation and chunked prefill. Table 1 provides benchmarking results of prefill–decode (PD) disaggregation. The takeaway is that (1) decode typically takes up the majority of energy consumption, and (2) PD disaggregation configurations do not have a noticeable impact on energy consumption, **as long as** prefill and decode throughputs are balanced. Tables 2 and 3 show benchmarking results of chunked prefill (energy and steady state power respectively), showing a very interesting non-linear impact of the max number of batched tokens, a parameter that controls the chunk size of chunked prefill.
>
> ---
> **Table 1\. Energy per generation (Joules) and the percentage of decode energy consumption with PD disaggregation.** Following recent trace analysis \[1\], we sampled input lengths from a Pareto distribution with alpha 2.5, and output lengths from an Exponential distribution, each with mean specified in the table.
>
> | Model and deployment \\ Request dataset | [Input 512, output 512] | [Input 512, output 4096] | [Input 4096, output 512] |
> | :---- | ----: | ----: | ----: |
> | Llama 3.1 8B (TP=1, 1P3D) | 37.71, 77.2% | 665.77, 98.7% | 208.34, 67.2% |
> | Llama 3.1 8B (TP=1, 2P2D) | 36.22, 76.7% | 706.27, 98.8% | 151.75, 55.2% |
> | Llama 3.1 8B (TP=1, 3P1D) | 37.26, 77.0% | 748.45, 98.9% | 158.85, 56.0%  |
> | Llama 3.1 70B (TP=4, 1P1D) | 276.93, 64.8% | 907.60, 89.2% | 1492.59, 50.0% |
> ---
>
> Decode generally consumes the larger part of total energy, with some amount shifting to prefill when input length is long. PD disaggregation configurations do not have a large impact on absolute energy consumption or the energy split as long as the throughput of prefill and decode instances are reasonably balanced (i.e., no specific prefill or decode instance limited in its running batch size). A notable imbalance case is the 8B model 1P3D deployment on long inputs, where the only prefill instance significantly bottlenecks decode instances and increases their energy consumption per generation by forcing them to run with a smaller batch size. Finally, when we go from 8B to 70B, prefill energy grows faster than decode. When we scale the model size, energy increase comes from both increased compute and memory operations. The 70B model consumes roughly 8.75 times more FLOPs for the compute-bound prefill and will increase prefill energy nearly proportionally. For decode, computation does increase but it is still not enough to saturate the device and the amount of KV cache only increases by 2.5 times, so the prefill energy grows faster than decode.
>
> ---
> **Table 2\. Energy per generation (Joules) of Llama 3.1 8B on long context requests and H100 GPUs.** Following recent trace analysis \[1\], we sampled the input lengths from a Pareto distribution with mean 4,096 and alpha 2.5, and output lengths from an Exponential distribution with mean 512. Please note that vLLM does not allow the max number of batched tokens to be smaller than the max batch size.
>
> | Max batched seqs \\ Max batched tokens | 32 | 64 | 128 | 256 | 512 | 1024 | 2048 | 4096 | 8192 |
> | :---- | ----: | ----: | ----: | ----: | ----: | ----: | ----: | ----: | ----: |
> | 32 | 559.66 | 374.63 | 269.29 | 205.54 | 188.80 | 191.61 | 195.59 | 191.88 | 194.52 |
> | 64 |  | 362.49 | 266.98 | 200.43 | 168.27 | 165.52 | 170.17 | 168.78 | 169.58 |
> | 128 |  |  | 264.18 | 194.59 | 164.75 | 154.64 | 155.59 | 156.54 | 156.93 |
> | 256 |  |  |  | 194.39 | 161.87 | 153.97 | 155.11 | 157.13 | 159.25 |
> | 512 |  |  |  |  | 159.57 | 151.50 | 154.52 | 156.77 | 154.95 |
> | 1024 |  |  |  |  |  | 152.67 | 156.26 | 157.98 | 163.08 |
> ---
>
> **Table 3\. Average steady state power draw (Watts) of Llama 3.1 8B Instruct on a synthetic long context request dataset running on H100 GPUs.** This is the same experiment as Table 2, but shows power instead of energy.
>
> | Max batched seqs \\ Max batched tokens | 32 | 64 | 128 | 256 | 512 | 1024 | 2048 | 4096 | 8192 |
> | :---- | ----: | ----: | ----: | ----: | ----: | ----: | ----: | ----: | ----: |
> | 32 | 483.77 | 570.32 | 620.14 | 684.07 | 640.20 | 619.54 | 618.39 | 602.34 | 607.65 |
> | 64 |  | 547.04 | 663.49 | 689.04 | 686.38 | 671.41 | 668.74 | 656.19 | 658.15 |
> | 128 |  |  | 654.32 | 690.39 | 685.44 | 689.94 | 683.71 | 680.25 | 677.27 |
> | 256 |  |  |  | 690.20 | 687.93 | 689.05 | 683.48 | 678.80 | 683.67 |
> | 512 |  |  |  |  | 688.50 | 689.06 | 682.72 | 682.04 | 678.49 |
> | 1024 |  |  |  |  |  | 689.86 | 687.78 | 678.04 | 681.03 |
> ---
>
> Table 2 shows that the more sequences or tokens you batch, the better the energy amortization you get and energy per generation decreases, and after a certain point, returns diminish. Table 3 is more interesting; as we decrease the max number of batched tokens while fixing the max batch size, power draw remains roughly the same, slowly increases, and then experiences a sudden drop (especially for smaller max batch sizes). As we slowly decrease the max number of batched tokens, the input prompt of a request can no longer fit into one batch’s token budget in full, and more and more iterations become chunked prefills piggybacking on decode. This increases the average arithmetic intensity of each iteration, which in turns increases power draw. Then, after some point, decreasing the max number of batched tokens reduces the total amount of work (computation) in a batch and ends up underutilizing the GPU’s compute resources. This leads to (1) lower power draw, which explains the sudden dip in power draw in Table 3, and (2) leads to the wastage of the static power draw of idle on-chip computing resources, which increases energy consumption per fixed amount of work, which is reflected by the big jump in energy per generation in Table 2.
>
> \[1\] Xiang et al. 2025\. ServeGen: Workload Characterization and Generation of Large Language Model Serving in Production.

---

> > ### Comment · Reviewer_c1Qn · 2025-08-01
> >
> > Thank you for the thorough ablation study. With these representative scenarios addressed, I believe the paper is now more complete. I also appreciate you sharing your experience with TPU and AMD GPU—very helpful context. Based on the updates provided in the rebuttal, I am inclined to raise my score.

---

> > > ### Author Response · Authors · 2025-08-01
> > >
> > > Thank you! Please let us know if we can provide additional context.

---

### Official Review · Reviewer_JN2P · 2025-07-05

**Rating:** 5
**Confidence:** 4

**Summary:**

The authors release the ML.Energy Benchmark, which provides a framework for evaluating the energy requirements of generative language, multimodal, and video workloads. The proposed benchmark supports the evaluation of a variety of serving framework and configuration design choices. The work provides additional analysis of factors, including model architecture and computational intensity, which affect the total energy required by the workload.

**Dataset Code Accessibility:**

Yes

**Ethical Considerations:**

No, there are no or only very minor ethics concerns

**Final Justification:**

See comment.

**Limitations Weaknesses:**

1. **Lack of Evaluations for Reasoning Tasks and Thinking Models**. As mentioned in Section 4.2 and in [3,4], energy use during LLM inference is highly dependent on output sequence length and verbosity.  Reasoning tasks require longer output generations for CoT thinking chains. The benchmark would benefit from evaluation with reasoning models and tasks (e.g. the inclusion of Qwen 3, DeepSeek Distil models; and GPQA, Math datasets).
2. **Lack of specification in the reported metrics**.
	1. How is GPU-power measured and what is the reported frequency? How is monetary cost determined? What is the PUE and carbon intensity used for CO2e determination?
	2. On examination of the web leaderboard, the additional metrics of CO2e and monetary costs specified in Sec 3.4 are not provided in reported results.

### Questions
1. What is the variation in steady state power draw for LLMs? for diffusion models?
	1. Does steady-state power draw vary for models of a single architecture? size?
2. As observed in [1] and in Appendix C, LLM prefill and decoding exhibit distinct power consumption profiles.
	1. To what extent do each of these stages contribute to the **total energy use**  for the processing of the entire request?
	2. How does this vary across examples with shorter vs longer input and output sequence lengths? Or with model size?
	3. Will power draw profiles and execution traces be provided in the released benchmark?
3. What are the set and values of hyperparameters explored in the search for pareto optimal configurations? What is the energy and emission cost of the search?
4. Will the benchmark support open submissions? Are the software configurations extensible (e.g. could vLLM be hot swapped for other inference engines such as TRT-LLM, SGLang, etc.)?


###  References
1. Patel, Pratyush, et al. "Characterizing power management opportunities for llms in the cloud." _Proceedings of the 29th ACM International Conference on Architectural Support for Programming Languages and Operating Systems, Volume 3_. 2024.
2. Li, Baolin, Yankai Jiang, and Devesh Tiwari. "Carbon in motion: Characterizing Open-Sora on the sustainability of generative AI for video generation." ACM SIGENERGY Energy Informatics Review 4.5 (2024): 160-165.
3. Samsi, Siddharth, et al. "From words to watts: Benchmarking the energy costs of large language model inference." _2023 IEEE High Performance Extreme Computing Conference (HPEC)_. IEEE, 2023.
4. Fernandez, Jared, et al. "Energy considerations of large language model inference and efficiency optimizations." arXiv preprint arXiv:2504.17674 (2025).
5. Jin, Yunho, Gu-Yeon Wei, and David Brooks. "The Energy Cost of Reasoning: Analyzing Energy Usage in LLMs with Test-time Compute." _arXiv preprint arXiv:2505.14733_ (2025).

**Strengths Contributions:**

1. **Support for wide variety of model architectures and multiple tasks.**  In contrast to prior evaluation benchmarks of energy and carbon emissions for ML (HuggingFace Energy Score, LLM-Perf), the proposed benchmark includes off-the-shelf support for a larger variety of serving configurations (e.g. parallelisms, batch sizes). As compared to prior analyses of LLM inference energy use, the benchmark provides greater coverage of task modalities [3,4,5].
2. **Analysis of Underlying Factors which determine the power utilization profile.** The authors characterize the relationships between stages in the computational inference workflow for LLMs and video generation models. To my knowledge, the characterization of power utilization and variability during the constituent stages of diffusion-based video generation is novel (Appendix C);  akin to the analysis of prefill vs decode power utilization in [1].

---

> ### Author Rebuttal · Authors · 2025-07-30
>
> We sincerely appreciate your feedback; it will improve not only the current submission, but also all future iterations of the benchmark.
>
> > Lack of Evaluations for Reasoning Tasks and Thinking Models.
>
> This is completely fair and we conducted experiments with different models on the GPQA dataset (Table 1). The takeaway is that reasoning models produce one to two orders of magnitude more output tokens per request, significantly increasing energy consumption per generation. Also, due to their large output length, servers cannot run as large a batch size, preventing them from amortizing energy across more requests, leading to higher energy per token, too.
>
> ---
> **Table 1\. Energy per generation (Joules) of reasoning models on GPQA and H100 GPUs.** Bold values indicate the max batch size that would be selected with a 50 ms target median inter-token latency. The server never reaches a running batch size of 64 due to long output lengths, so we stopped at 64\.
>
> | Model \\ Max batch size | 4 | 8 | 16 | 32 | 64 |
> | :---- | ----: | ----: | ----: | ----: | ----: |
> | DeepSeek distilled Qwen 3 8B (TP=1) | 9713.66 | 6010.10 | 4314.85 | 3340.75 | **2770.82** |
> | Phi 4 reasoning plus 15B (TP=1) | 19974.40 | 12389.63 | 9347.28 | **7634.94** | 7595.41 |
> | Qwen 3 32B (TP=2) | 26419.69 | 15168.31 | 9140.52 | 6165.50 | **4520.63** |
> | Qwen 3 235B-A22B thinking (TP=8) | 122523.05 | 86491.51 | **56720.41** | 40275.46 | 33096.43 |
> ---
>
> > Lack of specification in the reported metrics.
>
> We apologize for the lack of clarity in our submission; the intention of Section 3.4 (which mentioned average power draw, monetary cost, and operational carbon emissions) was to emphasize the usefulness of measuring energy by providing ways in which energy can be used to derive meaningful downstream metrics with very simple calculations, not to claim that these metrics are automatically reported by our benchmark.
>
> We did not make these additional metrics built-in by default mainly for two reasons.
>
> 1. We wanted the focus of the benchmark and leaderboard to be energy, instead of overwhelming users with too much information.
> 2. More importantly, deriving the value of these metrics is highly dependent on the context within which the benchmark was executed – especially the geographical location and time frame (which determines power mix, carbon intensity, and electricity price) and computing environment (which determines temperature, power delivery, and power usage effectiveness).
>
> Yet, we realize that the information in the current Section 3.4 may not be enough for non-expert users to derive those metrics based on energy. We will additionally discuss what is provided by the benchmark by default (energy, power, and time), how to obtain key pieces of information, and how to compute metrics. For instance, operational carbon intensity can be obtained from ElectricityMaps, electricity price from OpenEI, and specific datacenter PUEs from sources like the Google datacenter efficiency report. The frequency of power measurement is 10 times per second; NVIDIA GPUs do not support any finer-grained measurement at the time of writing.
>
> > What is the variation in steady state power draw for LLMs? for diffusion models? Does steady-state power draw vary for models of a single architecture? size?
>
> We have not seen cases where an LLM or a diffusion model’s power timeline **shape** deviates significantly from what is shown in Appendix C, Figure 12\. That is, LLMs show localized power peaks whenever there are (chunked) prefill iterations, troughs when there is CPU overhead momentarily underutilizing the GPU (e.g., scheduling, input processing, Python garbage collection), and generally draw lower power during decode iterations. On the other hand, diffusion models draw more constant power and typically stay close to the GPU’s max power.
>
> However, particularly for LLMs, the power timeline’s **level** (i.e., steady state power) can vary. Different model architectures and sizes can draw different power; this is shown in Appendix C, Figure 11\. Further, even for the same architecture, request dataset, and hardware, varying max batch size can affect steady state power draw (Table 2). The takeaway is that larger batch sizes generally increase steady state power draw. This is because the arithmetic intensity of LLM decode increases with batch size, which increases hardware utilization and thus power draw. We do note that higher batch sizes also increase the amount of communication between GPUs in the case of multi-GPU inference, which can dilute arithmetic intensity and lower power draw.
>
> ---
> **Table 2\. Average steady state power draw (Watts) of reasoning models on GPQA and H100 GPUs.** The server never reaches a running batch size of 64 due to long output lengths, so we stopped at 64\.
>
> | Model \\ Max batch size | 4 | 8 | 16 | 32 | 64 |
> | :---- | ----: | ----: | ----: | ----: | ----: |
> | DeepSeek distilled Qwen 3 8B (TP=1) | 403.26 | 439.56 | 493.05 | 552.48 | 598.66 |
> | Phi 4 reasoning plus 15B (TP=1) | 457.77 | 483.49 | 545.53 | 585.60 | 595.19 |
> | Qwen 3 32B (TP=2) | 905.54 | 928.10 | 968.39 | 1062.46 | 1171.78 |
> | Qwen 3 235B-A22B thinking (TP=8) | 2041.77 | 2098.92 | 2068.81 | 2097.74 | 2104.14 |
> ---
>
> > As observed in Patel et al. and in Appendix C, LLM prefill and decoding exhibit distinct power consumption profiles.
> > To what extent do each of these stages contribute to the total energy use for the processing of the entire request?
> > How does this vary across examples with shorter vs longer input and output sequence lengths? Or with model size?
>
> It is very difficult, technically, to separate prefill and decode energy consumption for two reasons.
>
> 1. The power counter resolution of GPUs is 100 ms, but these individual operations often take less. Essentially, GPU power counters today are only appropriate for measuring longer operations like the steady state.
> 2. With chunked prefill, which recent vLLM versions enable by default, partial prefills and decode operations happen simultaneously. However, all existing energy measurement tools provide only whole-device measurements. Separating and attributing whole-device energy to concurrent operations is itself a research problem.
>
> Instead, one way to see the split between prefill and decode is to benchmark prefill–decode (PD) disaggregation, which is a rising production deployment setting today. The takeaway is that (1) decode typically takes up the majority of energy consumption, and (2) PD disaggregation configurations do not have a noticeable impact on energy consumption, **as long as** prefill and decode throughputs are balanced. Due to space limitations, we ask that you refer to Table 1 in our response to Reviewer c1Qn for result data and more detailed explanation.
>
> > Will power draw profiles and execution traces be provided in the released benchmark?
>
> Authors are not allowed to update the repository during the rebuttal period, so we will open those upon acceptance.
>
> > What are the set and values of hyperparameters explored in the search for pareto optimal configurations? What is the energy and emission cost of the search?
>
> We first note that the search space is entirely configurable, and our exact search space can be found in YAML files in our GitHub repository.
>
> Batch size is the most important parameter. We did power-of-two batch sizes and intermediate average values. Particularly for LLMs, we stop increasing batch size when the server becomes overloaded and its running batch size fails to reach 90% of the max batch size configuration. With smaller models, shorter input/output lengths, and more GPU memory, the LLM server can scale to larger batch sizes. Another important factor is the number of GPUs to run the model, which is either one or two configurations primarily based on the size of the model (e.g., it rarely makes sense to serve an 8B model on four H100 80GB GPUs). The general rule is to ensure that model’ weights take up less than 50% of the total VRAM.
>
> We appreciate you asking about end-to-end cost. To be very honest, we never thought of measuring it. In future iterations of the benchmark, we hope to measure end-to-end costs.
>
> > Will the benchmark support open submissions?
>
> Open submissions should be verifiable by the maintainers for integrity and correctness. We are open to further discussion with interested parties.
>
> > Are the software configurations extensible?
>
> Yes. The LLM inference world has practically been unified with the OpenAI API schema, supported by all major inference servers. Therefore, all the user has to do is to extend the logic that spawns the server, and everything on top will work as is. Also, Zeus, the library we used to measure energy, supports not only PyTorch but also JAX, so we expect our benchmark to be extremely extensible.

---

> > ### Comment · Reviewer_JN2P · 2025-08-05
> >
> > Thanks to the author for their clarification and response -- I have raised my score. I ask that the additional results with reasoning models/tasks be included in the final revision; along with the methodology for determining non-energy cost metrics (potentially in appendix).

---

> > > ### Author Response · Authors · 2025-08-05
> > >
> > > Certainly! Thank you for your response, and please let us know if we can provide any additional context or clarification.

---

### Decision · Program_Chairs · 2025-09-18

**Decision:**

Accept (spotlight)

**Comment:**

--- This is still a draft, although the decision should not change ---

The authors propose a new benchmark to evaluate energy consumption (at inference time) of generative models. The benchmark allows to also optimise the energy consumption by finding the "best" model configuration for deployment.

There was a consensus among reviewers that the benchmark was quite carefully designed. Moreover, the optimisation part is very compelling and could be quite useful.

During the authors/reviewers discussion, the lack of evaluation of chain-of-though reasoning models was highlighted as a weakness, and the authors added new experiments using such models on the GPQA dataset (Rein et al., 2023). This addition is most welcome, and the authors are strongly encouraged to add it to the paper, and discuss it (the authors mention in the discussion that the energy footprint of reasoning is quite significant).

Given the quality of the paper, the addition of reasoning models, and the timeliness of the problem, I strongly believe that the paper should be accepted.
# Additional references

- Rein et al., GPQA: A Graduate-Level Google-Proof Q&A Benchmark, arXiv:2311.12022, 2023